# Genome-wide mapping of sister chromatid exchange events in single yeast cells using Strand-seq

Clémence Claussin[1], David Porubský[1], Diana CJ Spierings[1], Nancy Halsema[1], Stefan Rentas[2], Victor Guryev[1], Peter M Lansdorp[1,2,3]*, Michael Chang[1]*

[1]European Research Institute for the Biology of Ageing, University Medical Center Groningen, University of Groningen, Groningen, Netherlands; [2]Terry Fox Laboratory, BC Cancer Agency, Vancouver, Canada; [3]Department of Medical Genetics, University of British Columbia, Vancouver, Canada

**Abstract** Homologous recombination involving sister chromatids is the most accurate, and thus most frequently used, form of recombination-mediated DNA repair. Despite its importance, sister chromatid recombination is not easily studied because it does not result in a change in DNA sequence, making recombination between sister chromatids difficult to detect. We have previously developed a novel DNA template strand sequencing technique, called Strand-seq, that can be used to map sister chromatid exchange (SCE) events genome-wide in single cells. An increase in the rate of SCE is an indicator of elevated recombination activity and of genome instability, which is a hallmark of cancer. In this study, we have adapted Strand-seq to detect SCE in the yeast *Saccharomyces cerevisiae*. We provide the first quantifiable evidence that most spontaneous SCE events in wild-type cells are not due to the repair of DNA double-strand breaks.

DOI: https://doi.org/10.7554/eLife.30560.001

*For correspondence:
plansdor@bccrc.ca (PML);
m.chang@umcg.nl (MC)

**Competing interests:** The authors declare that no competing interests exist.

## Introduction

Homologous recombination (HR) is indispensable for the maintenance of genome integrity, and mutations in many HR genes are linked to a number of human diseases, especially various types of cancer (*Prakash et al., 2015*). HR is characterized as a major pathway to repair DNA double-strand breaks (DSBs), and can occur between any two homologous sequences, such as sister chromatids, homologous chromosomes, or homologous sequences located at different genomic loci (*Jasin and Rothstein, 2013*). HR results in a unidirectional transfer of genetic information from one DNA molecule to another (i.e. gene conversion) and/or a reciprocal exchange of genetic information between the two DNA molecules (i.e. a crossover). A crossover involving two homologous chromosomes can cause loss of heterozygosity, which is a common occurrence in cancer, while a crossover involving homologous sequences located at different genomic loci will result in chromosomal rearrangements, such as deletions, inversions, and translocations, which have been linked to a variety of human diseases (*Kong et al., 2011*). Thus, in mitotic cells, HR is biased towards noncrossovers, and HR involving identical sister chromatids, referred to as sister chromatid recombination (SCR), is preferred (*Kadyk and Hartwell, 1992*; *Nassif et al., 1994*; *Johnson and Jasin, 2000*; *Virgin et al., 2001*; *Stark and Jasin, 2003*), in part because a crossover during SCR should not lead to a change in DNA sequence.

However, studying SCR has been difficult, precisely because it does not normally result in a change in DNA sequence. Nevertheless, an SCR event that leads to a crossover, referred to as a sister chromatid exchange (SCE), can be detected. The mechanistic details of SCE and HR have been best studied using genetic assays in the budding yeast *Saccharomyces cerevisiae* (*Symington et al.,*

*2014*), though these assays are limited in several ways. First, perfect, or 'equal', chromosomal SCE events cannot be detected genetically, so most SCE assays measure 'unequal' SCE events, requiring recombination between misaligned sister chromatids. However, using a plasmid-based assay that creates a DSB on one of the two sister plasmids after DNA replication, equal SCE was found to occur much more frequently than unequal SCE (*González-Barrera et al., 2003*). Second, an often-used approach for detecting unequal SCE involves using two overlapping segments of a reporter gene in a tail-to-head configuration (*Fasullo and Davis, 1987*). The generation of a full-length reporter gene can result from an unequal SCE event, but it can also result from a long-tract gene conversion event, which does not involve a crossover. These two outcomes can be distinguished by analysis of sectored colonies (*Kadyk and Hartwell, 1992*), but this additional analysis is often not performed. Third, these assays measure recombination at a single genetic locus by introducing genetic markers at the locus. Introduction of such 'reporter cassettes' may disrupt the native chromatin environment, which could affect HR at the locus. Assaying recombination only at a single locus also ignores the rest of the genome. Some proteins important for HR may only affect specific regions of the genome, so these assays would not be useful to elucidate their functions. Lastly, most genetic assays rely on the outgrowth of a recombined cell. Thus, recombination events that lead to a reduction in cell proliferation capacity will be underrepresented, or left uncharacterized.

To alleviate these limitations, we previously developed a DNA template strand sequencing technique, called Strand-seq, which is capable of detecting SCE events genome-wide in single cells (*Falconer et al., 2012*). In Strand-seq, cells are allowed to divide once in the presence of bromodeoxyuridine (BrdU), a synthetic analogue of thymidine. BrdU gets incorporated into the newly synthesized DNA strands. Single daughter cells are isolated and the nascent DNA strands are selectively degraded to isolate parental template strands for the construction of directional sequencing libraries. Multiple single-cell libraries containing unique index sequences are pooled and sequenced on an Illumina platform. Aligned reads are binned into non-overlapping segments and plotted as coloured horizontal lines along an ideogram of each chromosome. An SCE is revealed by a switch in reads mapping to the Watson strand to reads mapping to the Crick strand, or vice versa (*Figure 1A*). Strand-seq was originally developed for use with mammalian cells. However, since most mammalian cells are diploid, it is not easy to differentiate between an SCE from an interhomolog recombination event. Yeast, on the other hand, can be studied in both haploid and diploid states, and the molecular details of HR are best known from studies in yeast. Thus, we have adapted the Strand-seq approach to work with yeast cells. We observe that 23% of wild-type cells have at least one SCE—the first time spontaneous SCEs have been measured and mapped genome-wide in yeast. Interestingly, our findings suggest that most spontaneous SCE events are not due to the repair of DSBs.

## Results

### Analyzing sister chromatid exchange in single yeast cells using Strand-seq

*S. cerevisiae* cells are unable to 'salvage' nucleosides from the environment, so to perform Strand-seq with yeast cells, we use yeast strains that express both *Drosophila melanogaster* deoxyribonucleoside kinase (dNK) and the human equilibrative nucleoside transporter (hENT1), allowing cells to take up and incorporate exogenous thymidine and BrdU into their DNA (*Vernis, 2003*). *CDC21*, which encodes thymidylate kinase, is also deleted in these strains, forcing them to be entirely dependent upon exogenous thymidine for viability. When grown in the presence of BrdU, these strains do not show any cell cycle delay or DNA damage checkpoint activation during the first cell cycle (*Vernis, 2003*). In addition, we see no significant difference in Rad52 focus formation, a sensitive indicator of recombination activity (*Lisby et al., 2001*), after 90 min of growth in BrdU-containing medium, suggesting that BrdU does not stimulate SCE (*Figure 1—figure supplement 1*). Although we cannot exclude the possibility that BrdU incorporation causes DNA damage that induces SCE in a manner that does not elicit a DNA damage checkpoint response or Rad52 focus formation, these findings are in line with our recently reported observation that BrdU incorporation also does not induce SCE in human cells, at least during the first two cell divisions, as varying the concentration of BrdU in the culture medium does not affect the frequency of SCE (*van Wietmarschen and Lansdorp, 2016*). We

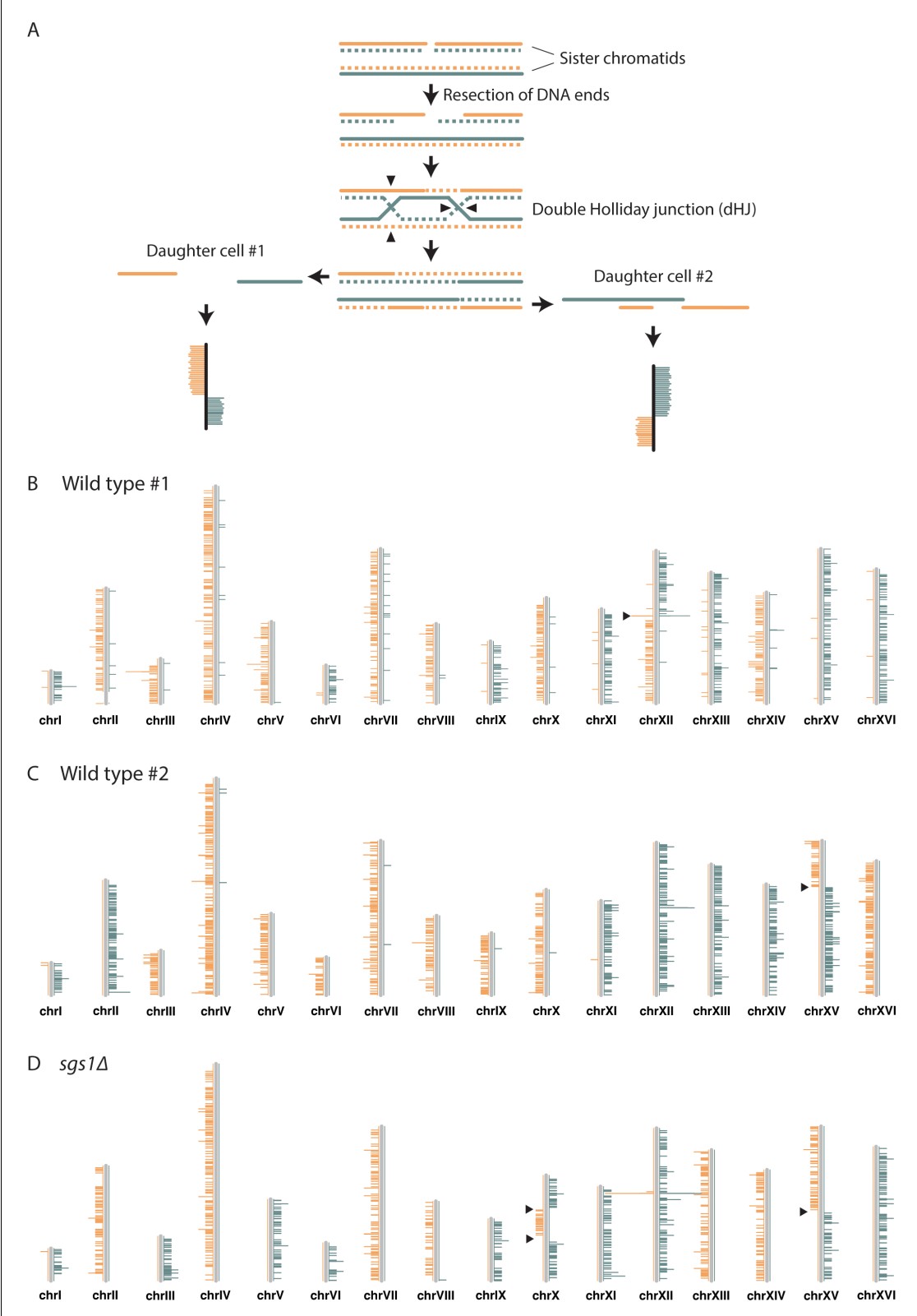

**Figure 1.** Detection of SCE events using Strand-seq. (**A**) An SCE can occur as a result of DSB repair. Two sister chromatids, one of which has a DSB, are shown. The parental template DNA strands are depicted with solid lines, while the newly synthesized strands containing BrdU are depicted with dashed lines. The Watson and Crick strands are shown in orange and blue, respectively. DSB repair by SCR can lead to the formation of a double Holliday junction (dHJ). Resolution of the dHJ by structure-specific endonucleases will result in either a noncrossover (not shown) or a crossover. The resulting
*Figure 1 continued on next page*

Figure 1 continued

sister chromatids are then segregated to two different daughter cells. In the current Strand-seq protocol, only one daughter cell is isolated and analyzed. The BrdU-containing strands are nicked during library preparation, resulting in the sequencing of only parental strands. Sequence reads are mapped to either side of a chromosome ideogram. An SCE results in a switch from Watson to Crick reads along the chromosome. Note: the small gap between the parental strands in daughter cell #1 and the small overlap of the parental strands in daughter cell #2 are too small to be detected with Strand-seq. (B) An example of a wild-type Strand-seq library. Ideograms of the 16 yeast chromosomes are shown. Orange and blue lines correspond to reads aligning to the Watson and Crick strands, respectively. This cell inherited either the parental Watson strand or the parental Crick strand for each chromosome, except chromosome XII. A switch from Watson to Crick reads can be seen for chromosome XII (black arrowhead), indicating that an SCE event has occurred. (C) A second example of a wild-type Strand-seq library. An SCE event was detected on chromosome XV. (D) An example of an *sgs1Δ* Strand-seq library. Three SCE events were detected in this library: two on chromosome X and one on chromosome XV.

DOI: https://doi.org/10.7554/eLife.30560.002

The following figure supplement is available for figure 1:

**Figure supplement 1.** Incorporation of BrdU does not increase Rad52 focus formation during the first cell cycle.

DOI: https://doi.org/10.7554/eLife.30560.003

release dNK- and hENT1-expressing *cdc21Δ* yeast cells synchronously from a G1 arrest into medium supplemented with BrdU and monitor cell cycle progression by flow cytometry. Cells that undergo precisely one cell division are isolated by cell sorting, followed by single cell Strand-seq library preparation (*Sanders et al., 2017*). Two examples of wild-type Strand-seq libraries, each containing one SCE, are shown in *Figure 1B and C*. We sequenced Strand-seq libraries derived from 218 wild-type cells and observed a total of 57 SCE events, which equates to 0.26 SCE events per cell (*Figure 2A*, *Table 1*). 23% of wild-type cells have at least one SCE (*Figure 2B*).

To validate our approach, we analyzed SCE in a hyperrecombinogenic strain (*sgs1Δ*), and in a strain lacking Rad52 (a key HR protein important for all HR-mediated activities). Mutation of BLM,

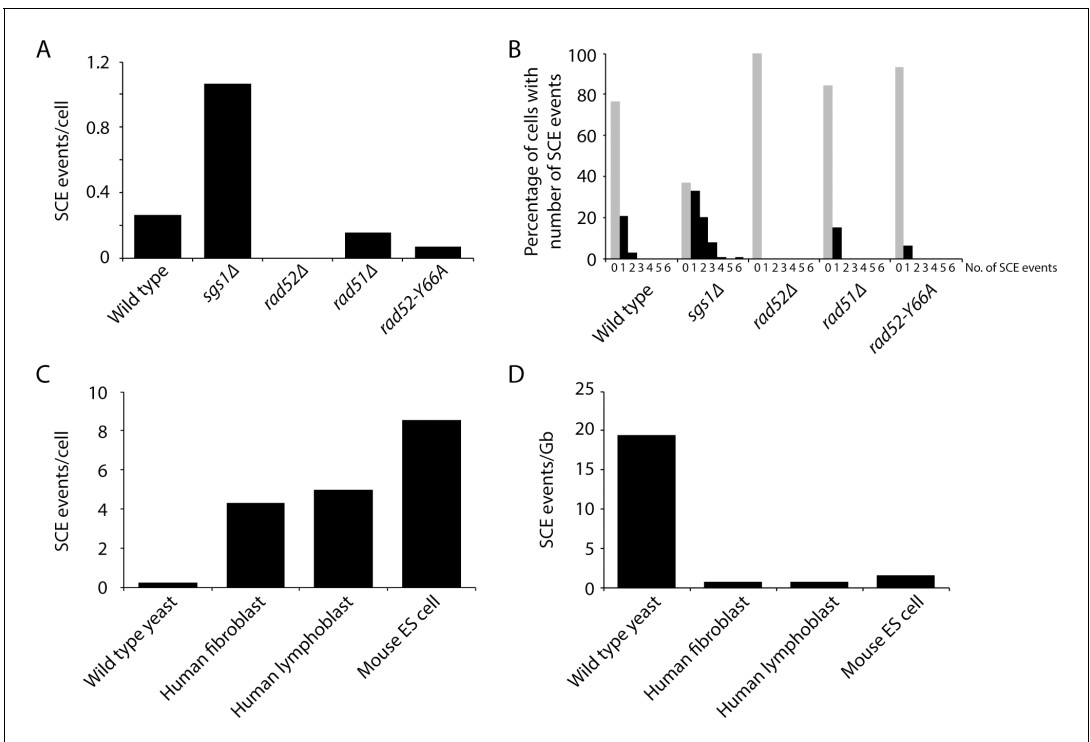

**Figure 2.** Measurement of spontaneous SCE genome-wide in single cells. (A) Number of SCE events per cell for the indicated genotypes. (B) Percentage of cells with the indicated number of SCE events for each genotype. (C) Number of SCE events per cell for wild-type yeast, human fibroblasts, human lymphoblasts, and mouse ES cells. (D) SCE events per gigabase of DNA for wild-type yeast, human fibroblasts, human lymphoblasts, and mouse ES cells.

DOI: https://doi.org/10.7554/eLife.30560.004

**Table 1.** Comparison of SCE by genotype.

| Genotype | No. of cells analyzed | No. of SCEs (SCEs/cell) | No. of SCEs at rDNA | SCEs/Gb (outside rDNA) | SCEs/Gb (within rDNA) |
|---|---|---|---|---|---|
| Wild type | 218 | 57 (0.26) | 14 | 16.3 | 47.0 |
| sgs1Δ | 103 | 110 (1.07) | 25 | 68.2 | 177.8 |
| rad52Δ | 27 | 0 (0) | 0 | 0 | 0 |
| rad51Δ | 65 | 10 (0.15) | 6 | 5.1 | 67.6 |
| rad52-Y66A | 76 | 5 (0.07) | 4 | 1.1 | 38.6 |

DOI: https://doi.org/10.7554/eLife.30560.005

the human homolog of *SGS1*, increases SCE (*German et al., 1965*; *Chaganti et al., 1974*; *van Wietmarschen and Lansdorp, 2016*), and deletion of *SGS1* has been reported to increase unequal SCE using two different genetic assays (*Onoda et al., 2000*; *Fasullo and Sun, 2017*). The evolutionarily conserved Sgs1-Top3-Rmi1 complex is thought to suppress SCE by dissolving a recombination intermediate known as the double Holliday junction (dHJ; see *Figure 1A*) in a manner that would result in a non-crossover (*Ira et al., 2003*; *Wu and Hickson, 2003*). Consistent with these findings, we observe an increase in SCE in *sgs1Δ* cells ($p < 10^{-9}$) using Strand-seq (*Figure 1D*; *Figure 2A and B*; *Table 1*). In addition, we did not detect any SCE events in *rad52Δ* cells, as expected (*Figure 2A and B*; *Table 1*).

The Strand-seq approach allows us to directly and quantitatively compare recombination activity in cells from yeast and mammals. Using previously reported Strand-seq data from mouse embryonic stem cells (*Falconer et al., 2012*) and human fibroblasts and lymphoblasts (*van Wietmarschen and Lansdorp, 2016*), we find that spontaneous SCE events per cell is about an order of magnitude greater in human and mouse cells than in wild-type yeast cells (*Figure 2C*). However, the human and mouse genomes are much larger than the yeast genome. By plotting the number of SCE events per gigabase of DNA, it is apparent that SCE activity is about 20-fold higher in yeast (*Figure 2D*). It is generally thought that yeast is more recombinogenic than mammals, but a direct and quantitative comparison has not been available until now.

## Sister chromatid exchange at the ribosomal DNA locus

Currently, we are only able to map SCE events to a resolution of approximately 30 kb (*Figure 3A*), which, combined with the modest number of SCE events we have detected so far (57 total from 218 wild-type Strand-seq libraries; *Table 1*), prevents us from mapping SCE events to specific genomic loci or features. However, multiple SCE events were seen at the ribosomal DNA (rDNA) locus, which is located on chromosome XII and consists of approximately 150 rDNA repeats, each of which is 9.1 kb in size (the SCE event in the Strand-seq library shown in *Figure 1B* maps to the rDNA locus). Since the repeats are identical in sequence, we cannot determine where within the ~1.4 Mb rDNA locus an SCE event occurs. Therefore, we only report whether or not an SCE has occurred somewhere within the rDNA locus. Of the 57 spontaneous SCE events we observed in our wild-type Strand-seq libraries, 14 were found at the rDNA locus (*Table 1*). We find that the rate of spontaneous SCE per gigabase of DNA in the rDNA locus is about threefold higher compared to the rest of the genome. This result was observed both in the presence and absence of Sgs1 (*Figure 3B*), and is likely due to the presence of the replication fork barrier within each rDNA repeat (*Labib and Hodgson, 2007*).

## The strand annealing activity of Rad52 is important for sister chromatid exchange

Rad52 has two main biochemical functions. First, Rad52 mediates the exchange of replication protein A (RPA) for Rad51 on single-stranded DNA (ssDNA) to promote Rad51-catalyzed strand invasion (*Sung, 1997*; *New et al., 1998*). Second, Rad52 facilitates the annealing of complementary RPA-coated ssDNA, a function that is independent of Rad51 (*Mortensen et al., 1996*; *Shinohara et al., 1998*; *Sugiyama et al., 1998*) and augmented by Rad59 (*Petukhova et al., 1999*; *Davis and Symington, 2001*; *Wu et al., 2006*). To determine which function of Rad52 is important for SCE, we performed Strand-seq on *rad51Δ* and *rad52-Y66A* mutant cells. *rad52-Y66A* belongs to a class of

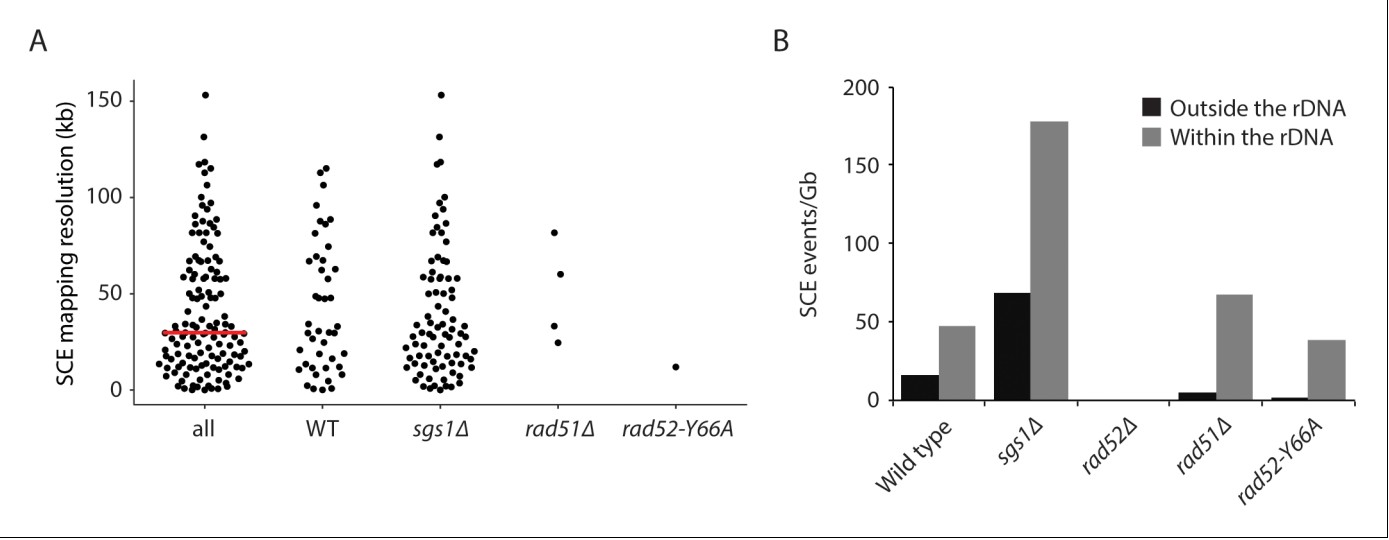

**Figure 3.** Mapping SCE events. (**A**) Mapping resolution of SCE events in all yeast Strand-seq libraries and by genotype. The red line shows the median mapping resolution for all libraries. SCE events within the rDNA locus are excluded from this analysis because it is not possible to determine where within the rDNA an SCE event has occurred. (**B**) SCE events per gigabase of DNA, for either the entire genome excluding the rDNA locus or only considering the rDNA locus, were plotted for the indicated genotypes.
DOI: https://doi.org/10.7554/eLife.30560.006

*rad52* mutants, class C mutants, that are mutated in the evolutionarily conserved N-terminal domain and share a similar phenotype of being proficient for mitotic recombination, but defective in the repair of DSBs (*Mortensen et al., 2002*). An in vitro study of one of the class C mutant proteins, rad52-R70A, indicated that these mutants are defective in strand annealing activity (*Shi et al., 2009*). We find that SCE is reduced 1.7-fold in *rad51Δ* cells and 4.5-fold in *rad52-Y66A* cells (p=0.058 and p<10$^{-4}$ for *rad51Δ* and *rad52-Y66A*, respectively; *Figure 2A and B*; *Table 1*), indicating that both functions of Rad52 are important for spontaneous SCE, with the strand annealing function being more so. This observation is in agreement with previous work indicating that the strand annealing activity of Rad52 is important for DSB-induced SCR (*Muñoz-Galván et al., 2013*). Using a genetic assay that detects unequal SCE, it was previously reported that Rad51 is not required for spontaneous SCE (*Fasullo et al., 2001*). However, this study did not assess the contribution of long-tract gene conversion in their assay, which may have obscured an actual decrease in unequal SCE in the *rad51Δ* mutant.

Interestingly, the effect of *rad51Δ* and *rad52-Y66A* on SCE is restricted to outside of the rDNA locus (3.2-fold [p=0.002] and 15-fold [p<10$^{-6}$] reduction, respectively, compared to wild type; *Figure 3B*). Within the rDNA locus, SCE in *rad51Δ* and *rad52-Y66A* is not significantly altered (p=0.48 and p=0.71, respectively). This observation is consistent with a previous study showing that spontaneous Holliday junction formation within the rDNA locus is dependent on Rad52, but not Rad51 (*Zou and Rothstein, 1997*). Taken together, the evidence suggests that Rad52 may have a function independent of Rad51 and strand annealing at the rDNA locus.

## Double-strand breaks are not the main cause of spontaneous sister chromatid exchange

It is often thought that SCE events are the result of the repair of DSBs (as depicted in *Figure 1A*). To investigate the role of DSBs in the generation of spontaneous SCE events in yeast, we compared the rate of spontaneous SCE in wild-type cells to the rate of DSB formation. Spontaneous DSB rate can be estimated by the rate of cell death in a *rad52Δ* mutant, which cannot repair DSBs (*Coïc et al., 2008*; *Mehta and Haber, 2014*). A single unrepaired DSB is sufficient to kill a *rad52* mutant cell (*Weiffenbach and Haber, 1981*). Thus, we used the mortality rate of our *rad52Δ* strain to estimate the rate of spontaneous DSB formation, which we found to be 12% per cell division (see Materials and methods). Using direct-repeat recombination assays to detect unequal SCR events, one quarter

of spontaneous (two of eight recombinants) and one third of X-ray-induced (four of 12 recombinants) SCR events were found to be accompanied by a crossover (*Jackson and Fink, 1981*; *Kadyk and Hartwell, 1992*). Therefore, if all spontaneous SCE events are the result of DSB repair, the rate of spontaneous SCE should be 3–4%. This rate represents an upper limit, because if Rad52 repairs other types of DNA damage that could lead to cell death if unrepaired, we would have overestimated the rate of spontaneous DSB formation. In contrast to this 3–4% upper limit, we find that 20% (43 out of 218, 95% CI [15%, 26%]) of wild-type Strand-seq libraries have at least one SCE not at the rDNA locus. We exclude SCE events that map to the rDNA locus because a DSB in the rDNA locus can be repaired by single-strand annealing in the absence of Rad52 due to the highly repetitive nature of the locus (*Ozenberger and Roeder, 1991*). Therefore, our results suggest that the majority of spontaneous SCE events are not the result of DSB repair.

Spontaneous HR initiated by non-DSB damage has been reported before, so we investigated whether the same lesion could be responsible for spontaneous SCE. For example, cells lacking both Sgs1 and the structure-specific endonuclease Mus81 are inviable, but this lethality can be suppressed by mutations in the *RAD52* epistasis group (i.e. *RAD51*, *RAD52*, *RAD54*, *RAD55*, and *RAD57*), indicating that *mus81Δ sgs1Δ* synthetic lethality is due to a toxic recombination intermediate (*Fabre et al., 2002*). Since Rad52 is required for the repair of DSBs, Fabre and colleagues argued that the toxic recombination intermediate cannot be initiated by a DSB. Similarly, cells lacking Top3 or Rmi1 grow very poorly, but this slow growth can be suppressed by mutations in *SGS1* or the members of the *RAD52* epistasis group, supporting a model where Rad52 and Sgs1 function in a pathway to repair a spontaneous non-DSB lesion, in the process creating a toxic recombination intermediate requiring Top3 and Rmi1 to resolve (*Gangloff et al., 1994*; *Shor et al., 2002*; *Chang et al., 2005*; *Mullen et al., 2005*). Unlike *rad51Δ* and *rad52Δ*, which can suppress the synthetic lethality of *mus81Δ sgs1Δ* and the slow growth of *rmi1Δ* (*Fabre et al., 2002*; *Chang et al., 2005*), we find that *rad52* class C mutants, which are defective in Rad52-mediated strand annealing (*Shi et al., 2009*) and spontaneous SCE (*Figure 2A and B*), cannot suppress either phenotype (*Figure 4*). These findings indicate that the DNA lesion(s) that cause *mus81Δ sgs1Δ* synthetic lethality and *rmi1Δ* slow growth are different than the DNA lesion(s) that cause spontaneous SCE in wild-type cells, and that HR is important to repair multiple types of endogenous DNA damage.

## Discussion

In this study, we have used Strand-seq to measure SCE genome-wide in individual yeast cells. We provide evidence that the majority of spontaneous SCE events in wild-type cells are not the result of DSB repair. Non-DSB lesions have also been invoked to explain the synthetic lethality of *mus81Δ sgs1Δ* and *sgs1Δ srs2Δ* mutants (*Fabre et al., 2002*) and the proficiency of *rad52* class C mutants for spontaneous inter- and intrachromosomal heteroallelic HR (*Lettier et al., 2006*). While the strand annealing activity of Rad52 is important for spontaneous SCE, it is not in the other noted instances of spontaneous HR, indicating that there must be at least two types of endogenous non-DSB lesions processed by HR. These lesions could be single-stranded nicks or gaps, as proposed in many of the original models for HR (*Holliday, 1964*; *Meselson and Radding, 1975*; *Radding, 1982*), or other type of structures interfering with strand-specific replication (*Figure 5A and B*). Indeed, it is well known that single-stranded gaps can initiate recombination in *Escherichia coli* (*Persky and Lovett, 2008*) and that fork collapse at the polar replication fork barrier *RTS1* in *Schizosaccharomyces pombe* induces HR-dependent replication restart from a single-stranded gap (*Lambert et al., 2010*).

The importance of strand annealing for spontaneous SCE is reminiscent of recombination involving inverted repeats, which requires Rad52 but is only modestly reduced by deletion of *RAD51* (*Rattray and Symington, 1994*). Rad59, which augments the strand annealing function of Rad52 (*Petukhova et al., 1999*; *Davis and Symington, 2001*; *Wu et al., 2006*), is required for spontaneous recombination of inverted repeats in the absence of Rad51 (*Bai and Symington, 1996*). It has been proposed that these events occur by template switching during DNA replication in a manner involving strand annealing (*Mott and Symington, 2011*). Poly-ubiquitylation of PCNA by the Mms2-Ubc13-Rad5 ubiquitin conjugating enzyme complex promotes repair through the error-free post-replication repair pathway, which is thought to involve template switching (*Xu et al., 2015*). Rad5 and Rad18 were found to have only a minor role in inverted-repeat recombination (*Mott and Symington, 2011*), and the rates of spontaneous SCE in *mms2Δ*, *ubc13Δ*, and *rad5Δ* mutants are

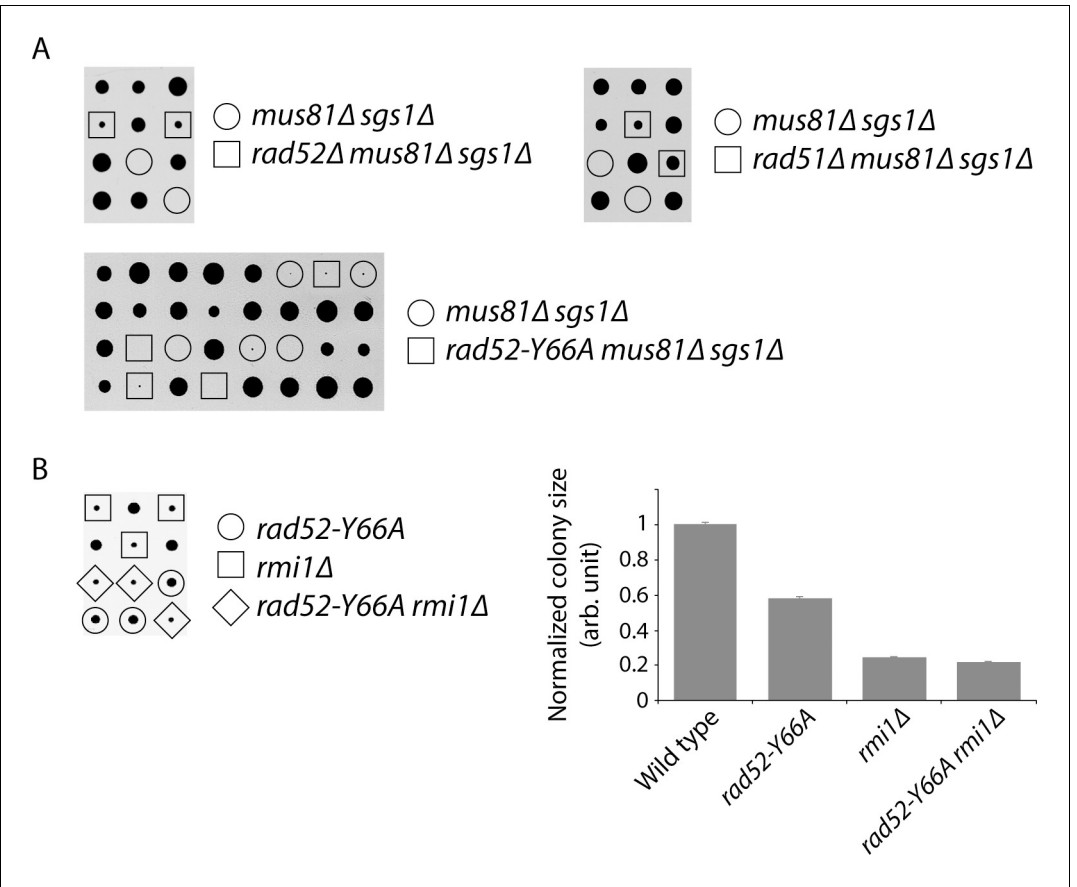

**Figure 4.** Abolishing the strand annealing activity of Rad52 does not suppress *mus81Δ sgs1Δ* synthetic lethality or *rmi1Δ* slow growth. (**A**) Representative tetrads derived from the sporulation of MCY736, MCY737, and MCY773 are shown. (**B**) Representative tetrads derived from the sporulation of CCY198 are shown. Colony sizes for the indicated genotypes were measured and normalized to wild type. Mean ±SEM is shown. Lack of suppression of *mus81Δ sgs1Δ* synthetic lethality and *rmi1Δ* slow growth was also observed using another *rad52* class C mutant: *rad52-R70A* (data not shown).

DOI: https://doi.org/10.7554/eLife.30560.007

similar to wild type, as measured by an unequal SCE assay (*Fasullo and Sun, 2017*). However, template switching has been proposed to occur differently depending on whether the DNA lesion is on the leading strand template or the lagging strand template, with lesion bypass on the leading strand utilizing Rad5 while lesion bypass on the lagging strand mediated by Rad52 (*Gangavarapu et al., 2007*). Aside from the rDNA locus, we have so far been unable to map spontaneous SCE events to specific genomic features so further work will be needed to determine whether inverted repeats replicated by the lagging strand machinery are a source of spontaneous SCE (*Figure 5C*).

We were able to map multiple spontaneous SCE events to the rDNA locus because of its large size (approximately 10% of the genome). Due to its repetitive nature, the rDNA locus has been an excellent substrate for studying HR. Recombination within the locus is important to maintain repeat homogeneity and copy number homeostasis (*Kobayashi, 2011*). Consistent with the importance of HR at the rDNA locus, we observe that SCE is threefold higher at the rDNA locus compared to the rest of the genome (*Figure 3B*). Nevertheless, an SCE event occurs at the rDNA locus in only 6.4% (14 out of 218) of wild-type cells (*Table 1*). Considering that it has been estimated that a single cell has on average 3.6 Holliday junctions and one DSB within the rDNA array per cell cycle (*Zou and Rothstein, 1997*; *Sasaki and Kobayashi, 2017*), our data indicate that the vast majority of HR events at the rDNA locus do not result in crossovers, in agreement with previous work showing that HR at the rDNA locus is largely rearrangement-free (*Zou and Rothstein, 1997*).

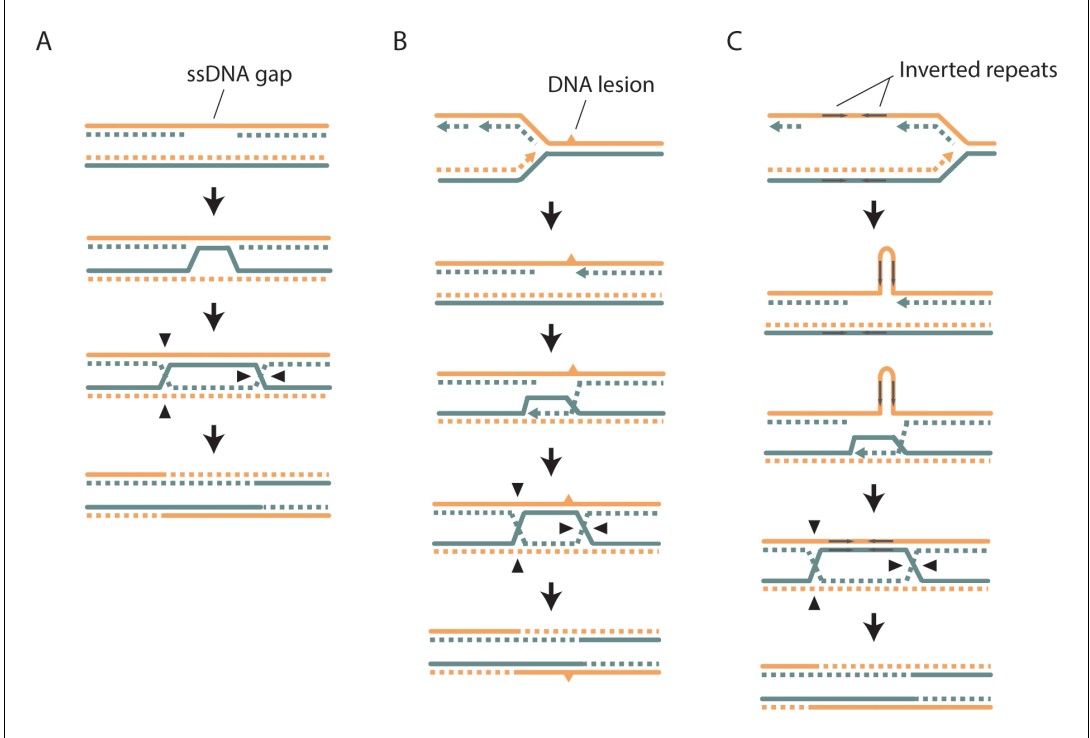

**Figure 5.** Models of SCE that do not involve a DSB. As in *Figure 1A*, the parental template DNA strands are depicted with solid lines, while the newly synthesized strands are depicted with dashed lines. The Watson and Crick strands are shown in orange and blue, respectively. (**A**) An SCE could be generated from the repair of an ssDNA gap. This could proceed via inverse strand exchange, where Rad52 forms a complex with dsDNA and promotes strand exchange with a homologous ssDNA sequence independently of Rad51 (*Mazina et al., 2017*). (**B**) A DNA lesion on one of the parental template strands can cause template switching, where nascent DNA is used as a template for DNA replication, and could result in SCE. (**C**) As in **B**), but the 'DNA lesion' is an inverted DNA repeat forming a hairpin loop.

DOI: https://doi.org/10.7554/eLife.30560.008

In addition to SCE, Strand-seq could in theory be used to detect break-induced replication (BIR) events involving sister chromatids. BIR occurs when one end of a DSB invades an intact homologous DNA sequence to generate a replication fork that can potentially proceed to the end of the chromosome. Unlike canonical DNA replication, which is semi-conservative, BIR occurs by conservative DNA synthesis (*Donnianni and Symington, 2013*; *Saini et al., 2013*). With Strand-seq, a chromosomal region synthesized by BIR could either have no Watson or Crick reads (i.e. both Watson and Crick strands are newly synthesized) or both Watson and Crick reads (i.e. both Watson and Crick strands consist of the original parental template strands). Application of Strand-seq in yeast will be particularly advantageous to address questions about various types of DNA lesions and replication stress-inducing barriers at specific locations in the genome. These and other potential applications of Strand-seq offer intriguing new possibilities to study HR, complimenting more traditional assays to yield a better picture of HR activity genome-wide at the level of single cells.

## Materials and methods

### Yeast strains and growth conditions

Strains used in this study are listed in *Table 2* and are all *RAD5* derivatives of W303 (*Thomas and Rothstein, 1989*; *Zhao et al., 1998*). The E17 strain was derived from YLV11 (*Vernis, 2003*). Unless noted otherwise, standard culturing conditions and genetic techniques were used to manipulate yeast strains (*Sherman, 2002*).

**Table 2.** Yeast strains used in this study.

| Strain name | Relevant genotype | Source |
|---|---|---|
| E17 | MATa ADE2 cdc21::kanMX leu2::LEU2-GAL-hENT1 LYS2 RAD5 trp1::TRP1-GAL-dNK ura3-1 | Peter Thorpe |
| CCY232 | MATa ADE2 cdc21::kanMX leu2::LEU2-GAL-hENT1 LYS2 RAD5 trp1::TRP1-GAL-dNK ura3-1 RAD52-GFP::HIS3M × 6 | This study |
| CCY234 | MATa ADE2 can1-100 his3-11,15 leu2-3,112 trp1-1 ura3-1 RAD5 RAD52-GFP::HIS3M × 6 | This study |
| CCY118 | MATa ADE2 cdc21::natMX leu2::LEU2-GAL-hENT1 LYS2 RAD5 trp1::TRP1-GAL-dNK ura3-1 sgs1ΔkanMX | This study |
| CCY193 | MATa ADE2 cdc21::natMX leu2::LEU2-GAL-hENT1 LYS2 RAD5 trp1::TRP1-GAL-dNK ura3-1 rad52ΔkanMX | This study |
| CCY150 | MATa ADE2 cdc21::kanMX leu2::LEU2-GAL-hENT1 LYS2 RAD5 trp1::TRP1-GAL-dNK ura3-1 rad51ΔnatMX | This study |
| CCY182 | MATa ADE2 cdc21::kanMX leu2::LEU2-GAL-hENT1 LYS2 RAD5 trp1::TRP1-GAL-dNK ura3-1 rad52-Y66A | This study |
| MCY736 | MATa/MATα ade2-1/ade2-1 can1-100/can1-100 his3-11,15/his3-11,15 leu2-3,112/leu2-3,112 trp1-1/trp1-1 ura3-1/ura3-1 RAD5/RAD5 mus81ΔkanMX/MUS81 rad52Δ/RAD52 sgs1ΔHIS3/SGS1 | This study |
| MCY737 | MATa/MATα ade2-1/ade2-1 can1-100/can1-100 his3-11,15/his3-11,15 leu2-3,112/leu2-3,112 trp1-1/trp1-1 ura3-1/ura3-1 RAD5/RAD5 mus81ΔkanMX/MUS81 rad51ΔnatMX/RAD51 sgs1ΔHIS3/SGS1 | This study |
| MCY773 | MATa/MATα ade2-1/ade2-1 can1-100/can1-100 his3-11,15/his3-11,15 leu2-3,112/leu2-3,112 trp1-1/trp1-1 ura3-1/ura3-1 RAD5/RAD5 rad52-Y66A::hphMX/RAD52 sgs1ΔHIS3/SGS1 | This study |
| CCY198 | MATa/MATα ade2-1/ade2-1 can1-100/can1-100 his3-11,15/his3-11,15 leu2-3,112/leu2-3,112 trp1-1/trp1-1 ura3-1/ura3-1 RAD5/RAD5 rad52-Y66A/RAD52 rmi1ΔkanMX/RMI1 | This study |
| MCY735 | MATa/MATα ade2-1/ade2-1 can1-100/can1-100 his3-11,15/his3-11,15 leu2-3,112/leu2-3,112 trp1-1/trp1-1 ura3-1/ura3-1 RAD5/RAD5 mus81ΔkanMX/MUS81 rad52-R70A/RAD52 sgs1ΔHIS3/SGS1 | This study |
| CCY196 | MATa/MATα ade2-1/ade2-1 can1-100/can1-100 his3-11,15/his3-11,15 leu2-3,112/leu2-3,112 trp1-1/trp1-1 ura3-1/ura3-1 RAD5/RAD5 rad52-R70A/RAD52 rmi1ΔkanMX/RMI1 | This study |

DOI: https://doi.org/10.7554/eLife.30560.009

## Isolation of single cells for Strand-seq

Strand-seq compatible strains (*cdc21::kanMX leu2::LEU2-GAL-hENT1 trp1::TRP1-GAL-dNK*) were grown in 25 ml of YP medium (1% yeast extract, 2% peptone) supplemented with 2% galactose (YPGal) and 100 μM thymidine to an $OD_{600}$ of 0.2–0.4. Cells were arrested in G1 phase by the addition of alpha mating factor (Sigma) for 2 hr. At this point, two different protocols were used: the first involved sorting single cells, while the second involved sorting individual spheroplasts. Both protocols yielded high quality Strand-seq libraries, with no change in the number of SCE events observed per cell (*Figure 6A*), nor in SCE mapping resolution (*Figure 6B*), but the spheroplast approach proved to be more reliable with less background sequencing reads. The Strand-seq library shown in *Figure 1B* was derived from the cell sorting approach, while the Strand-seq libraries shown in *Figure 1C and D* were derived from the spheroplast sorting approach.

For the cell sorting approach, G1-arrested cells were washed with water and resuspended in 100 μl of PBS solution. 150 μl of 200 μg/ml of Concanavalin A (ConA), conjugated to either Cy5 or Alexa Fluor 633, was then added, followed by a 10 min incubation at room temperature in the dark. ConA binds to the yeast cell wall (*Tkacz et al., 1971*). Unbound ConA was removed by washing three times with PBS. Cells were then synchronously released from G1 arrest into fresh YPGal supplemented with 0.6 mg/ml of BrdU. Samples were taken before G1 arrest, after G1 arrest, after ConA-labeling, and every 30 min after release from G1 arrest. The samples were fixed with 70% ethanol and stored for at least one night at −20°C. Cells were pelleted, resuspended in 0.5 ml of 50 mM Tris-Cl pH 8.0 plus 10 μl of 10 mg/ml RNase A, and incubated for 2–4 hr at 37°C. The cells were then washed once with 50 mM Tris-Cl pH 7.5, resuspended in 300 μl of 50 mM Tris-Cl pH 7.5, 1.5 μM SYTOX Green, and incubated overnight at 4°C. Cell suspensions were briefly sonicated and analyzed by flow cytometry. Newly divided daughter cells can be identified by having 1C DNA content and without ConA labeling, since yeast daughter cells bud off from the original mother cells, synthesizing a new cell wall that is not stained with ConA (*Chung et al., 1965*; *Figure 7*). The single daughter cells were then sorted with a MoFlo Astrios cell sorter (Beckman Coulter) into separate wells of a 96-well plate for Strand-seq library preparation.

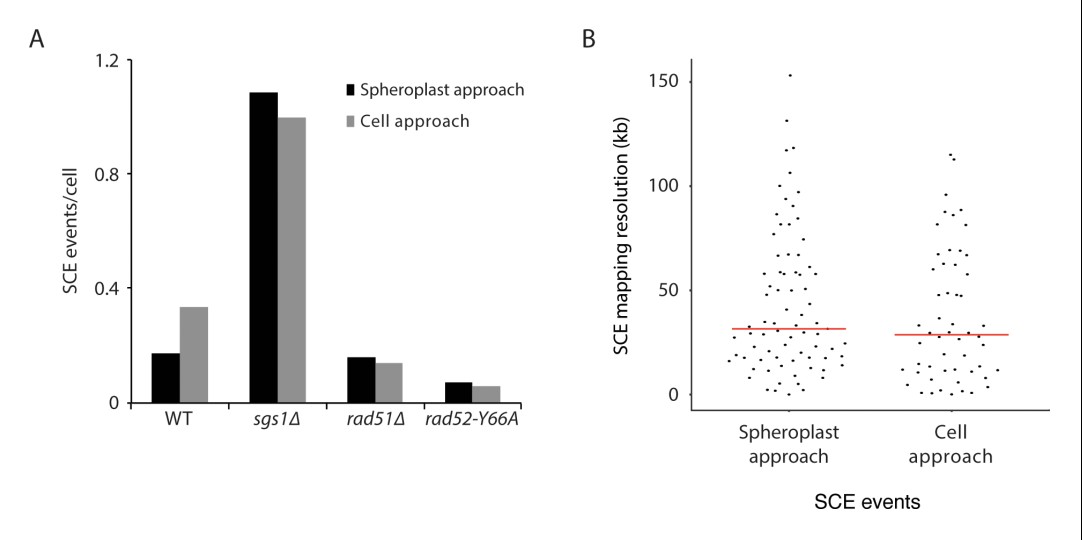

**Figure 6.** The spheroplast-sorting approach compared to the cell-sorting approach in terms of (**A**) number of SCE events per cell (divided by genotype) and (**B**) SCE mapping resolution.

DOI: https://doi.org/10.7554/eLife.30560.010

For the spheroplast sorting approach, G1 arrest and release into BrdU-containing media was performed as in the cell sorting protocol, with the omission of ConA-labeling. Cell cycle progression was monitored by flow cytometry. Cells that had progressed through one cell cycle were incubated 0.17 mg/ml zymolyase for 1 hr at 37°C. The resulting spheroplasts were washed in PBS and those with 1C DNA content were sorted with a FACSJazz cell sorter (BD Biosciences) into separate wells of a 96-well plate for Strand-seq library preparation. Without the ConA-labeling, we cannot distinguish newly divided cells from cells that remain arrested in G1 without having progressed through the cell cycle. However, the arrested cells will not have incorporated BrdU, resulting in Strand-seq libraries that will have both Watson and Crick reads for every chromosome. These libraries can be easily identified and discarded. Moreover, we find that the occurrence of such libraries is very rare.

## Strand-seq library preparation and sequencing

Strand-seq libraries were prepared as previously described (*Sanders et al., 2017*), with a few modifications. Sorted single yeast cells were first treated with zymolyase, followed by incubation with MNase to fragment the DNA. Several zymolyase reaction conditions were used in an attempt to optimize the procedure. In the end, we found that 0.175 mg/ml zymolyase for 30 min at 30°C worked best. Sorted single spheroplasts were directly subjected to MNase treatment. Adapter ligation was performed in the presence of PEG 6000, using the Quick Ligation Reaction Buffer (New England Biolabs) and 10 nM adapters to promote efficient ligation of adapters to DNA templates and reduce adapter dimer formation. After ligation and PCR, DNA clean-up with magnetic AMPure XP beads (Agencourt AMPure, Beckman Coulter) was performed twice using a 1.2 vol of beads. Single-end 50 bp sequencing reads from the Strand-seq libraries were generated using the HiSeq 2500 or the NextSeq 500 sequencing platform (Illumina, San Diego, CA). Reads were aligned to the *S. cerevisiae* EF4 reference genome using Bowtie2 (version 2.0.0-beta7; *Langmead and Salzberg, 2012*) and analyzed using the BAIT software package (*Hills et al., 2013*) for data selection and visualization purposes. Only libraries with an average read count greater than 20 per megabase of DNA were included in the analyses. SCE events were detected using an R-based package called HapSCElocatoR (publicly available through GitHub: https://github.com/daewoooo/HapSCElocatoR; *Porubský and Sanders, 2017* ; copy archived at https://github.com/elifesciences-publications/HapSCElocatoR). HapSCElocatoR uses circular binary segmentation (implemented in the R package fastseg; *Klambauer et al., 2012*) to localize SCE events in haploid Strand-seq libraries as a change in read directionality from Watson (negative strand) to Crick (positive strand) or vice versa. Only non-duplicate reads with a mapping quality greater than or equal to 10 were analyzed. We considered

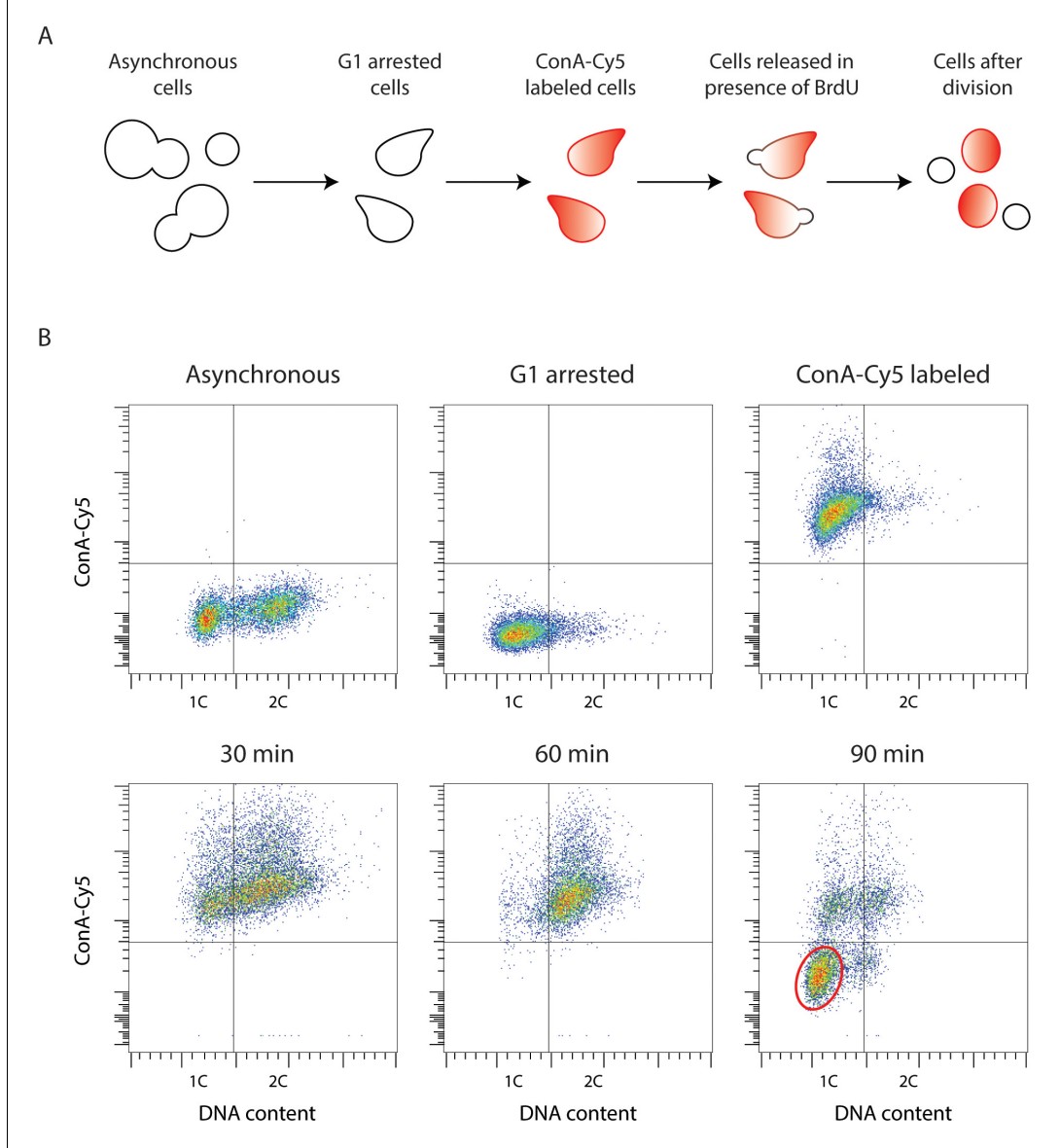

**Figure 7.** Yeast Strand-seq via the 'single cell sorting' approach. (A) Scheme of the 'single cell sorting' approach for yeast Strand-seq. (B) Representative flow cytometry scatter plots corresponding to the scheme depicted in (A). Cells appearing in the lower left quadrant in the 90 min post-release sample are newly divided daughter cells to be sorted. A typical window for sorting is shown by the red oval.

DOI: https://doi.org/10.7554/eLife.30560.011

only strand state changes with at least three directional reads on both sides of the putative SCE site as an SCE event. Single directional reads embedded within an extended region with the opposite directionality were considered as errors and their directionality was flipped. Computationally localized SCE events were further manually verified by visual inspection of chromosome ideograms (obtained from BAIT; see *Figure 1B*). Comparisons of SCE rate between different genotypes were assessed for statistical significance using a *t*-test.

## Live cell imaging of cells expressing Rad52-GFP

Strains CCY232 and CCY234 were inoculated in the morning in 2 mL of synthetic medium supplemented with 100 μM thymidine and grown at 25°C. In the evening, the cultures were diluted into 50 mL of synthetic medium supplemented with thymidine to reach an $OD_{600}$ of 0.2–0.4 in the following morning. Cells were then washed twice with synthetic medium and placed either into 25 mL of

synthetic medium supplemented with 100 μM thymidine (for CCY232 and CCY234) or 25 mL of synthetic medium supplemented with 200 μM BrdU (for CCY232) and grown for 90 min at 25°C—instead of 30°C, to allow the GFP chromophore to fold more efficiently (*Lim et al., 1995*). Cells were then mounted onto agar pads for imaging, which was performed using a Zeiss LSM 800 or a DeltaVision Image Restoration Microscope. Images were scores for cells with or without foci by two people independently.

## Estimation of spontaneous DSB rate

Freshly grown E17 (wild type) or CCY193 (*rad52Δ*) were spread onto one side of a YPGal +100 μM thymidine plate. Cells were micromanipulated to designated locations on the plate and allowed to divide once. If a cell failed to divide at this stage, it was discarded from further analysis. This step ensured that we would look at healthy and newly divided cells. From the cells that do divide, each daughter cell was then micromanipulated away from its mother cell to a new location on the same plate, which was then incubated for 2–3 days at 30°C to allow the isolated mother and daughter cells to form colonies. The number of mother or daughter cells that were unable to form a colony was recorded. Incidences where both the mother and the daughter did not form a colony were excluded from the analysis because these mother-daughter pairs may have resulted from the division of an arrested cell with unrepaired DNA damage as a result of DNA damage checkpoint adaptation (*Clémenson and Marsolier-Kergoat, 2009*). In total, 408 E17 and 338 CCY193 cells were examined (i.e. 204 E17 and 169 CCY193 mother-daughter pairs). 13.3% (95% CI [9.9%, 17.4%]) of CCY193 cells and 1.7% (95% CI [0.7%, 3.5%]) of E17 cells failed to form a colony. E17 mortality is unlikely due to unrepaired DSBs, and possibly due to death caused by the micromanipulation process, so subtracting this 1.7% 'background' mortality rate from the 13.3% mortality rate of CCY193 cells yields an estimated spontaneous DSB rate of 11.6%. This mortality rate is similar to the 8.3% previously reported for *rad52Δ* of a different strain background (*Ozenberger and Roeder, 1991*).

## Acknowledgements

We thank Phil Hieter and Jan Stoepel (Michael Smith Laboratory, UBC, Vancouver), and the UMCG Central Flowcytometry Unit, for technical assistance with the sorting of yeast daughter cells; Peter Thorpe for providing the E17 strain; Niek van Wietmarschen, Karina Hoekstra-Wakker, and Jorn Staal for technical assistance with Strand-seq library preparation; and Peter Stirling and Rodney Rothstein for constructive comments on the manuscript. Financial support was provided by a European Research Council Advanced grant (to PML) and a Vidi grant from the Netherlands Organisation for Scientific Research (to MC).

## Additional information

### Funding

| Funder | Grant reference number | Author |
| --- | --- | --- |
| European Research Council | Advanced Grant | Peter M Lansdorp |
| Netherlands Organisation for Scientific Research | Vidi Grant 016.131.314 | Michael Chang |

The funders had no role in study design, data collection and interpretation, or the decision to submit the work for publication.

### Author contributions

Clémence Claussin, Conceptualization, Investigation, Methodology, Writing—original draft, Writing—review and editing; David Porubský, Data curation, Software, Writing—review and editing; Diana CJ Spierings, Investigation, Methodology, Writing—review and editing; Nancy Halsema, Investigation, Methodology; Stefan Rentas, Methodology, Writing—review and editing; Victor Guryev, Data curation, Software, Supervision, Writing—review and editing; Peter M Lansdorp, Conceptualization, Supervision, Funding acquisition, Methodology, Writing—review and editing; Michael Chang,

Conceptualization, Supervision, Funding acquisition, Investigation, Methodology, Writing—original draft, Project administration, Writing—review and editing

### Author ORCIDs
Clémence Claussin http://orcid.org/0000-0002-0994-3596
Peter M Lansdorp https://orcid.org/0000-0001-7435-1071
Michael Chang http://orcid.org/0000-0002-1706-3337

### Decision letter and Author response
Decision letter https://doi.org/10.7554/eLife.30560.014
Author response https://doi.org/10.7554/eLife.30560.015

## Additional files

### Supplementary files
• Transparent reporting form
DOI: https://doi.org/10.7554/eLife.30560.012

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
