## [Decision Letter]

Thank you for sending your article entitled "Double-strand breaks are not the main cause of spontaneous sister chromatid exchange in wild-type yeast cells" for peer review at *eLife*. Your article has been favorably evaluated by Jessica Tyler (Senior Editor) and three reviewers, one of whom is a member of our Board of Reviewing Editors.

The reviewers recognize the importance of sister chromatid exchange and the potential of Strand-seq to measure the number and distribution of these events genome wide. The two main concerns identified by the reviewers are: (1) could the high frequency of SCEs result from the method used to detect them; and (2) validity of the argument regarding the source of the initiating lesion of SCEs.

Essentially, the concern is that the conditions used to incorporate BrdU into the genome are non-physiological and could result in DNA damage that then leads to sister chromatid exchange. Reviewer 2 suggests specific experiments that could be used to address this concern. In particular, determining viability of rad52 cells under the same conditions used for Strand-seq. Activation of the DNA damage and/or replication checkpoint could also be assessed under these conditions.

*Reviewer #1:*

In this study, the authors apply a method called Strand-seq to determine the number and distribution of spontaneous sister chromatid exchanges (SCE) in the yeast, *Saccharomyces cerevisiae*. The advantage of this method over previously described assays in yeast, is that it measures events genome wide in a single generation, and reports on equal sister chromatid recombination in contrast to genetic assays that are usually designed to detect unequal recombination between repeated gene segments at a single locus. Yeast is an ideal system for Strand-seq because of the stable haploid state, which avoids complications due to inter-homolog recombination, and plethora of mutants available to investigate the role of genes in known DNA repair pathways in regulating sister chromatid recombination.

The authors found that 23% of wild type daughter cells analyzed had at least one SCE and these events were dependent on RAD52, and only partially dependent on RAD51. Interestingly, the *rad52-Y66A* allele, anticipated to be defective in the strand annealing function of Rad52, exhibited a greater decrease in SCEs outside of the rDNA than the *rad51* mutant. As expected, the number of spontaneous SCEs was significantly increased in the *sgs1/blm* mutant. The authors then speculate on the initiating lesion for SCEs and from indirect assays infer that most of the events must initiate from a lesion other than a double-strand break (DSB). While I agree with the view that most SCEs probably arise from a single-stranded gap formed during replication, I don't think this is such an unusual standpoint. Most of the *E. coli* recombination community is well aware of the importance of recombination initiated at gaps, and fission yeast labs studying recombination at stalled replication forks have long argued that the events studied do not go through a DSB intermediate. Furthermore, the 2002 study by Fabre and colleagues (cited here) made a strong case that most spontaneous recombination must initiate from ssDNA gaps. I suggest the authors tone done the section stating it is generally assumed recombination initiates from DSBs.

Subsection “The strand annealing activity of Rad52 is important for sister chromatid exchange”, last paragraph: I found the section on Rad52 requirement for SCE in the rDNA confusing. If Rad52 is required for Holliday junction formation in the rDNA, but not Rad51, it would suggest the Rad52 strand annealing activity is important. Yet *rad52-Y66A*, which is supposedly defective for the strand annealing function is proficient for rDNA recombination.

Figure 4: The *rad52* null should also be shown.

*Reviewer #2:*

Claussin et al. apply a novel and fairly sensitive assay for crossing-over between sister chromatids (a.k.a. SCE) to the scoring of inter-sister crossovers in single budding yeast cells. They provide evidence that these crossovers occur in about 1 in 4 cell cycles, and that most occur within the ribosomal DNA. They also show that these intersister crossovers require Rad52, with a partition between rDNA and single copy sequences in terms of the dependence on Rad51 and on the annealing activity of Rad52. They also claim to show, based on death rates in *rad52Δ* cells, and on the failure of *rad52* class C mutants (which lack annealing activity) to suppress the synthetic lethality of *mus81 sgs1* double mutants and the slow growth of *rmi1* mutants. This paper is interesting because it provides the first detection of intersister exchange in "normal" genomes, and provides evidence in further support of a partition between the rDNA repeats and the rest of the genome, in terms of recombination biochemistry. However, there is some question as to whether the frequencies of intersister crossovers reflect what happens in normal cells, and the argument regarding the relationship between double strand breaks and intersister crossovers is invalid.

1) Assay conditions might impact the frequency of recombinogenic lesions. In the strand-seq assay, cells are shifted from thymidine-containing medium into medium where BrdU is the only source of "T". Under these conditions, there are many non-physiological sources for induced damage. These include inefficient uptake or inefficient phosphorylation leading to decreased dNTP pools and subsequent fork stalling, recognition of incorporated BrU as a damaged base, removal by BER and subsequent post-replicative repair by recombination, or photolytic damage followed by repair by recombination. These are, of course, quite difficult to control for, and all might increase the observed frequency of intersister crossing-over beyond what occurs in the absence of BrdU. The only remedy that I can think of would be to incorporate one of the previously published assays for unequal sister chromatid exchange into the strains used, and compare rates of exchange genetically in cells grown in thymidine, cells subjected to BrdU as for the strand-seq protocol, and in otherwise isogenic CDC21 (i.e. wild-type) cells whose growth is independent of exogenous thymidine.

2) Cell inviability assays compare rates of death in cells grown in thymidine with rates of SCE in cells grown in BrdU. Because of uncertainty regarding whether or not BrdU induces additional SCE in yeast (see point above), the argument comparing SCE rates with death rates is currently invalid.

3) While a single DSB is sufficient to kill a *rad52* mutant cells, this mostly occurs by irreversible DNA damage checkpoint arrest. It is not known whether or not a DSB in one of two sister chromatids is also sufficient to cause irreversible arrest. If it is, then these cells would not divide, and thus would be excluded from the spontaneous DSB rate assay, because they would not have divided. This would lead to an underestimate of lethality, and thus the frequency of unrepaired lethal lesions.

In sum, a better assay for cell viability would take the same population used for strand-seq, and to plate cells after S is completed, but before the first mitotic division, on YPD + thymidine, and then scoring for both cells that fail to divide, and for cells where either mother or daughter are unable to form colonies.

4) Failure of *rad52Δ* or *rad52-Y66A* to suppress the lethality of a *mus81 sgs1* double mutant, or the slow growth of an *rmi1* mutant does not mean that the lesions responsible for the synthetic lethal or slow growth phenotypes are not double-strand breaks, just that Rad52 is not required for these lesions to form intermediates that require Mus81 or Sgs1-Top3-Rmi1 activity for resolution.

The comments above should not be taken as suggesting that *only* double-strand breaks induce mitotic recombination, just that the evidence the authors have put forward is insufficient to prove their argument.

*Reviewer #3:*

Sister-chromatid exchange (SCE) is one of the major sources of genome rearrangement. However, SCE without rearrangement (equal SCE), that is thought to occur most frequently, has not been analyzed because of the identical sequence after recombination. To reveal the mechanism of SCE and the resulting genome instability, it is important to know the nature of equal SCE, too.

The authors applied the Strand-seq method to yeast and mapped 57 SCE sites in analyzed 218 cells. 14 out of 57 SCEs occurred in the rDNA that is known as a hot spot of rearrangement (copy number variation). Interestingly, the authors speculated that most of the SCEs are not the results of DSB repair because frequency of the repair is estimated much less than that of SCE from the information of previous reports. The authors are presenting three models that make SCE possible without DSB repair.

The conclusion is interesting and valuable to understand the actual mechanism of SCE. However, I think some descriptions in the paper are inappropriate. The authors should revise the following points before publication.

1) Subsection “Sister chromatid exchange at the ribosomal DNA locus”: HOT1 is thought to be an element that works outside of the rDNA by the function of pol I transcription (Huang and Keil, 1995: Ward et al., 2000; Serizawa et al., 2004, etc.). The recombination in the rDNA mainly depends on the replication fork barrier (RFB) activity by Fob1. The author should rewrite the sentence.

2) The authors discuss only about equal SCE. I wonder if they didn't find any rearrangement that really increases genome instability. They should add the information if they have.

3) What causes the SCE? Replication fork inhibition is thought to be the main trigger. I think it may be worth doing to test the effect of a replication inhibitor to the rate of SCE. This experiment may support the model but not essential for the conclusion of this paper.

4) Is there any hotspot of SCE? The authors should discuss about the sequences or regions in which SCE is induced.

[Editors' note: further revisions were requested prior to acceptance, as described below.]

Thank you for resubmitting your work entitled "Double-strand breaks are not the main cause of spontaneous sister chromatid exchange in wild-type yeast cells" for further consideration at *eLife*. Your revised article has been favorably evaluated by Jessica Tyler (Senior Editor) and three reviewers, one of whom is a member of our Board of Reviewing Editors.

The manuscript has been improved but there are some remaining issues that need to be addressed before acceptance, as outlined below:

Reviewers 1 and 3 are satisfied with the revision, but reviewer 2 is unconvinced that BrdU is not causing some damage that fails to elicit a checkpoint response or Rad52 foci. We don't really know what a Rad52 focus denotes, other than it is some kind of DNA damage. Rothstein's group has claimed, in fact, that multiple lesions are incorporated into a single Rad52 focus. Clearly cells with SCEs exceed cells with Rad52 foci by at least a factor of 2, and given observations that non-crossovers are formed in excess over crossovers in mitotic cells, it seems clear that most of the recombinogenic lesions formed are not detected as Rad52 foci. None of the assays the current manuscript contains directly address the question of whether or not spontaneous DSBs or SCEs occur at the same frequencies when cells are grown in thymidine versus BrdU. Ideally, one would want to measure spontaneous USCE using a genetic assay under the same conditions used here, one cycle of growth with BrdU. However, as noted in the authors rebuttal, there are technical issues with doing that type of analysis. All reviewers agree that there are a number of interesting findings presented, but given the concern over whether BrdU could cause damage and induce SCEs we suggest that you modify the title, something along the lines of "Direct measurement of sister chromatid exchange in single cells", and tone done the suggestion that DSBs are not the cause of spontaneous SCEs.

---

## [Author Response]

Reviewer #1:In this study, the authors apply a method called Strand-seq to determine the number and distribution of spontaneous sister chromatid exchanges (SCE) in the yeast, Saccharomyces cerevisiae. The advantage of this method over previously described assays in yeast, is that it measures events genome wide in a single generation, and reports on equal sister chromatid recombination in contrast to genetic assays that are usually designed to detect unequal recombination between repeated gene segments at a single locus. Yeast is an ideal system for Strand-seq because of the stable haploid state, which avoids complications due to inter-homolog recombination, and plethora of mutants available to investigate the role of genes in known DNA repair pathways in regulating sister chromatid recombination.The authors found that 23% of wild type daughter cells analyzed had at least one SCE and these events were dependent on RAD52, and only partially dependent on RAD51. Interestingly, the rad52-Y66A allele, anticipated to be defective in the strand annealing function of Rad52, exhibited a greater decrease in SCEs outside of the rDNA than the rad51 mutant. As expected, the number of spontaneous SCEs was significantly increased in the sgs1/blm mutant. The authors then speculate on the initiating lesion for SCEs and from indirect assays infer that most of the events must initiate from a lesion other than a double-strand break (DSB). While I agree with the view that most SCEs probably arise from a single-stranded gap formed during replication, I don't think this is such an unusual standpoint. Most of the E. coli recombination community is well aware of the importance of recombination initiated at gaps, and fission yeast labs studying recombination at stalled replication forks have long argued that the events studied do not go through a DSB intermediate. Furthermore, the 2002 study by Fabre and colleagues (cited here) made a strong case that most spontaneous recombination must initiate from ssDNA gaps. I suggest the authors tone done the section stating it is generally assumed recombination initiates from DSBs.

In the revised manuscript, we have removed instances where we state that it is generally assumed spontaneous SCE events are initiated by DSBs. Also, we have cited the relevant *E. coli* and fission yeast studies (see first paragraph of the Discussion).

Subsection “The strand annealing activity of Rad52 is important for sister chromatid exchange”, last paragraph: I found the section on Rad52 requirement for SCE in the rDNA confusing. If Rad52 is required for Holliday junction formation in the rDNA, but not Rad51, it would suggest the Rad52 strand annealing activity is important. Yet rad52-Y66A, which is supposedly defective for the strand annealing function is proficient for rDNA recombination.

We agree with the reviewer that this section needs more clarity. We find that neither Rad51 nor Rad52-mediated strand annealing are important for SCE at the rDNA locus. Our observation is consistent with that of Zou and Rothstein, who found that neither Rad51 nor Rad59 are important for the formation of spontaneous Holliday junctions at the rDNA (as we noted in our manuscript). Together, the evidence suggests that Rad52 may have a function independent of Rad51 and strand annealing.

We have now revised the text accordingly.

Figure 4: The rad52 null should also be shown.

Rescue of *rmi1∆* slow growth by *rad52∆* has been previously reported (Chang et al., 2005).

This reference has now been included.

Reviewer #2:Claussin et al. apply a novel and fairly sensitive assay for crossing-over between sister chromatids (a.k.a. SCE) to the scoring of inter-sister crossovers in single budding yeast cells. They provide evidence that these crossovers occur in about 1 in 4 cell cycles, and that most occur within the ribosomal DNA. They also show that these intersister crossovers require Rad52, with a partition between rDNA and single copy sequences in terms of the dependence on Rad51 and on the annealing activity of Rad52. They also claim to show, based on death rates in rad52Δ cells, and on the failure of rad52 class C mutants (which lack annealing activity) to suppress the synthetic lethality of mus81 sgs1 double mutants and the slow growth of rmi1 mutants. This paper is interesting because it provides the first detection of intersister exchange in "normal" genomes, and provides evidence in further support of a partition between the rDNA repeats and the rest of the genome, in terms of recombination biochemistry. However, there is some question as to whether the frequencies of intersister crossovers reflect what happens in normal cells, and the argument regarding the relationship between double strand breaks and intersister crossovers is invalid.1) Assay conditions might impact the frequency of recombinogenic lesions. In the strand-seq assay, cells are shifted from thymidine-containing medium into medium where BrdU is the only source of "T". Under these conditions, there are many non-physiological sources for induced damage. These include inefficient uptake or inefficient phosphorylation leading to decreased dNTP pools and subsequent fork stalling, recognition of incorporated BrU as a damaged base, removal by BER and subsequent post-replicative repair by recombination, or photolytic damage followed by repair by recombination. These are, of course, quite difficult to control for, and all might increase the observed frequency of intersister crossing-over beyond what occurs in the absence of BrdU. The only remedy that I can think of would be to incorporate one of the previously published assays for unequal sister chromatid exchange into the strains used, and compare rates of exchange genetically in cells grown in thymidine, cells subjected to BrdU as for the strand-seq protocol, and in otherwise isogenic CDC21 (i.e. wild-type) cells whose growth is independent of exogenous thymidine.

We have now measured Rad52 focus formation in the Strand-seq strain in the presence and absence of BrdU and found no significant difference (see new Figure 1—figure supplement 1). We also measured Rad52 focus formation in a wild-type *CDC21* strain and found that focus formation is similar to what has been previously reported by Lisby et al., 2001 (PMID: 11459964).

2) Cell inviability assays compare rates of death in cells grown in thymidine with rates of SCE in cells grown in BrdU. Because of uncertainty regarding whether or not BrdU induces additional SCE in yeast (see point above), the argument comparing SCE rates with death rates is currently invalid.

See above.

3) While a single DSB is sufficient to kill a rad52 mutant cells, this mostly occurs by irreversible DNA damage checkpoint arrest. It is not known whether or not a DSB in one of two sister chromatids is also sufficient to cause irreversible arrest. If it is, then these cells would not divide, and thus would be excluded from the spontaneous DSB rate assay, because they would not have divided. This would lead to an underestimate of lethality, and thus the frequency of unrepaired lethal lesions.In sum, a better assay for cell viability would take the same population used for strand-seq, and to plate cells after S is completed, but before the first mitotic division, on YPD + thymidine, and then scoring for both cells that fail to divide, and for cells where either mother or daughter are unable to form colonies.

We believe that an unrepaired DSB on one of the two sister chromatids would indeed be sufficient to cause an irreversible arrest in *rad52∆* cells, and the assay we used is designed to take this into account. We believe the confusion lies in how we described the assay. The goal of the assay is to determine the percentage of *rad52∆* cells that die because of damage occurring during a single cell division. We could have isolated individual cells and asked how many of them could form a colony. However, if done this way, we would have included cells that may have inherited irreparable damage from the previous cell cycle (e.g. they were the products of checkpoint adaptation). This would have inflated the mortality rate. Thus, we instead dissected mother-daughter pairs to ensure that we were looking at healthy cells that had just divided. If a mother or daughter then acquires an irreparable DSB (because *RAD52* is deleted), it will not form a colony and this would be recorded. We apologize for the confusion and have modified the text to clarify this point.

4) Failure of rad52Δ or rad52-Y66A to suppress the lethality of a mus81 sgs1 double mutant, or the slow growth of an rmi1 mutant does not mean that the lesions responsible for the synthetic lethal or slow growth phenotypes are not double-strand breaks, just that Rad52 is not required for these lesions to form intermediates that require Mus81 or Sgs1-Top3-Rmi1 activity for resolution.The comments above should not be taken as suggesting that only double-strand breaks induce mitotic recombination, just that the evidence the authors have put forward is insufficient to prove their argument.

To clarify, *rad52∆*does suppress the lethality of *mus81∆ sgs1∆* and the slow growth of *rmi1∆*, but *rad52-Y66A*does not. Thus, Rad52 (but not its strand annealing activity) is required for the formation of the intermediates that require Mus81 or Sgs1-Top3-Rmi1 activity for resolution. Rad52 is required for DSB repair, strongly suggesting that the spontaneous lesions cannot be DSBs. This was the conclusion drawn by Fabre and colleagues, as mentioned by reviewer #1 and cited in our manuscript.

Reviewer #3:Sister-chromatid exchange (SCE) is one of the major sources of genome rearrangement. However, SCE without rearrangement (equal SCE), that is thought to occur most frequently, has not been analyzed because of the identical sequence after recombination. To reveal the mechanism of SCE and the resulting genome instability, it is important to know the nature of equal SCE, too.The authors applied the Strand-seq method to yeast and mapped 57 SCE sites in analyzed 218 cells. 14 out of 57 SCEs occurred in the rDNA that is known as a hot spot of rearrangement (copy number variation). Interestingly, the authors speculated that most of the SCEs are not the results of DSB repair because frequency of the repair is estimated much less than that of SCE from the information of previous reports. The authors are presenting three models that make SCE possible without DSB repair.The conclusion is interesting and valuable to understand the actual mechanism of SCE. However, I think some descriptions in the paper are inappropriate. The authors should revise the following points before publication.1) Subsection “Sister chromatid exchange at the ribosomal DNA locus”: HOT1 is thought to be an element that works outside of the rDNA by the function of pol I transcription (Huang and Keil, 1995: Ward et al., 2000; Serizawa et al., 2004, etc.). The recombination in the rDNA mainly depends on the replication fork barrier (RFB) activity by Fob1. The author should rewrite the sentence.

We now refer to the replication fork barrier rather than HOT1.

2) The authors discuss only about equal SCE. I wonder if they didn't find any rearrangement that really increases genome instability. They should add the information if they have.

We indeed found evidence for other types of rearrangements. However, we currently do not have enough information to draw any conclusions.

3) What causes the SCE? Replication fork inhibition is thought to be the main trigger. I think it may be worth doing to test the effect of a replication inhibitor to the rate of SCE. This experiment may support the model but not essential for the conclusion of this paper.

Using genetic unequal SCE assays, it has been reported by others (e.g. work from Mike Fasullo’s lab) that replication inhibitors and DNA damaging agents stimulate SCE. However, this does not conclusively indicate that replication fork inhibition is the cause of *spontaneous* SCE events. Nonetheless, it is an interesting idea, and we do plan to test in the future the effect of replication inhibitors with Strand-seq, but we believe this is outside the scope of the present study.

4) Is there any hotspot of SCE? The authors should discuss about the sequences or regions in which SCE is induced.

As discussed in our manuscript, we have not been able to identify hotspots of SCE (beyond the rDNA locus) because (i) we do not have enough SCE events from our Strand-seq libraries and (ii) the resolution in mapping these events is currently insufficient to determine hotspots.

[Editors' note: further revisions were requested prior to acceptance, as described below.]

The manuscript has been improved but there are some remaining issues that need to be addressed before acceptance, as outlined below:Reviewers 1 and 3 are satisfied with the revision, but reviewer 2 is unconvinced that BrdU is not causing some damage that fails to elicit a checkpoint response or Rad52 foci. We don't really know what a Rad52 focus denotes, other than it is some kind of DNA damage. Rothstein's group has claimed, in fact, that multiple lesions are incorporated into a single Rad52 focus. Clearly cells with SCEs exceed cells with Rad52 foci by at least a factor of 2, and given observations that non-crossovers are formed in excess over crossovers in mitotic cells, it seems clear that most of the recombinogenic lesions formed are not detected as Rad52 foci. None of the assays the current manuscript contains directly address the question of whether or not spontaneous DSBs or SCEs occur at the same frequencies when cells are grown in thymidine versus BrdU. Ideally, one would want to measure spontaneous USCE using a genetic assay under the same conditions used here, one cycle of growth with BrdU. However, as noted in the authors rebuttal, there are technical issues with doing that type of analysis. All reviewers agree that there are a number of interesting findings presented, but given the concern over whether BrdU could cause damage and induce SCEs we suggest that you modify the title, something along the lines of "Direct measurement of sister chromatid exchange in single cells", and tone done the suggestion that DSBs are not the cause of spontaneous SCEs.

We are happy to hear that our revised manuscript has been favorably evaluated by the senior and reviewing editors, along with the three reviewers. As suggested by the reviewers, we have now changed the title to “Genome-wide mapping of sister chromatid exchange events in single yeast cells using Strand-seq”. We also note that our experiment assessing the percentage of cells that exhibit a Rad52 focus after 90 minutes of growth in the presence of either BrdU or thymidine does *not* give an accurate representation of the percentage of cells that form a Rad52 focus during a single cell cycle. Our experiment only provides a snapshot at one specific point during the cell cycle for each cell. The Rothstein and Lisby labs have performed time-lapse experiments to follow Rad52 focus formation throughout the cell cycle in 10-minute intervals (see PMID: 17096599, 17591698, 21075850). They have found that ~50% of cells form a Rad52 focus per cell cycle. This is likely an underestimate because short-lived foci may have been missed between the time points. Therefore, our observed value for 3 the percentage of cells with an SCE is very much in line with the reported percentage of cells that form a Rad52 focus per cell cycle.